# Effect of Physical Activity on Cardiovascular Event Risk in a Population-Based Cohort of Patients with Type 2 Diabetes

**DOI:** 10.3390/ijerph182312370

**Published:** 2021-11-24

**Authors:** Mónica Enguita-Germán, Ibai Tamayo, Arkaitz Galbete, Julián Librero, Koldo Cambra, Berta Ibáñez-Beroiz

**Affiliations:** 1Unidad de Metodología, Navarrabiomed-HUN-UPNA, 31008 Pamplona, Spain; monica.enguita.german@navarra.es (M.E.-G.); ibai.tamayo.rodriguez@navarra.es (I.T.); arkaitz.galbete@unavarra.es (A.G.); julian.librero.lopez@navarra.es (J.L.); 2Instituto de Investigación Sanitaria de Navarra (IdiSNA), 31008 Pamplona, Spain; 3Red de Investigación en Servicios Sanitarios y Enfermedades Crónicas (REDISSEC), 48902 Bilbao, Spain; k-cambra@euskadi.eus; 4Departamento de Estadística, Universidad Pública de Navarra (UPNA), 31008 Pamplona, Spain; 5Departamento de Salud, Gobierno Vasco, 01006 Vitoria-Gasteiz, Spain

**Keywords:** cardiovascular disease, mortality, type 2 diabetes, physical activity, population-based cohort, nested case-control

## Abstract

Cardiovascular disease (CVD) is the most common cause of morbidity and mortality among patients with type 2 diabetes (T2D). Physical activity (PA) is one of the few modifiable factors that can reduce this risk. The aim of this study was to estimate to what extent PA can contribute to reducing CVD risk and all-cause mortality in patients with T2D. Information from a population-based cohort including 26,587 patients with T2D from the Navarre Health System who were followed for five years was gathered from electronic clinical records. Multivariate Cox regression models were fitted to estimate the effect of PA on CVD risk and all-cause mortality, and the approach was complemented using conditional logistic regression models within a matched nested case–control design. A total of 5111 (19.2%) patients died during follow-up, which corresponds to 37.8% of the inactive group, 23.9% of the partially active group and 12.4% of the active group. CVD events occurred in 2362 (8.9%) patients, which corresponds to 11.6%, 10.1% and 7.6% of these groups. Compared with patients in the inactive group, and after matching and adjusting for confounders, the OR of having a CVD event was 0.84 (95% CI: 0.66–1.07) for the partially active group and 0.71 (95% CI: 0.56–0.91) for the active group. A slightly more pronounced gradient was obtained when focused on all-cause mortality, with ORs equal to 0.72 (95% CI: 0.61–0.85) and 0.50 (95% CI: 0.42–0.59), respectively. This study provides further evidence that physically active patients with T2D may have a reduced risk of CVD-related complications and all-cause mortality.

## 1. Introduction

Diabetes mellitus is a highly prevalent metabolic disorder that has become an undeniable health problem in recent decades, especially in developed countries, with cardiovascular disease (CVD) being the most common cause of morbidity and mortality among these patients [1]. Efforts are needed to cost-effectively prevent or at least delay the appearance of such CVD-related events and to reduce all-cause mortality in this at-risk population. Physical activity (PA) is an easily modifiable lifestyle factor that is widely recommended due to its demonstrated beneficial effect on overall health. Both recreational and nonrecreational moderate and intense PA have been associated with lower mortality and CVD risk in studies conducted in the general population [2,3]. Furthermore, moderate and vigorous PA reduces the association between sedentary behavior and CVD and cancer-related mortality [4].

People with type 2 diabetes (T2D) have specific features that lead them to higher CVD risk compared with the general population; therefore, the effect of protective or risk factors on both populations may be different. In fact, some CVD risk models derived for the general population have been shown to be suboptimal when applied to patients with T2D [5,6]. In patients with type 2 diabetes (T2D), different exercise modalities, such as walking, aerobic, resistance, strength, yoga or certain combinations, have been found to improve the levels of the most important CVD risk factors, including hemoglobin A1c (HbA1c), glucose, low-density lipoprotein (LDL), total cholesterol, heart rate and weight [7,8,9,10,11,12,13,14]. Most of the evidence relating to PA and CVD risk is based on short-term interventional randomized clinical trials and focused on these intermediate outcomes. Although evidence on sustained PA effects on final endpoints within prospective cohorts and population-based studies using real-world data has increased in the last decade [15], studies conducted specifically on patients with T2D are still limited [16,17,18]. In fact, very few models that predict CVD risk include this important modifiable risk factor as a covariate [19], despite the suggested importance of its protective effect [15] and the clinical practical guides’ recommendations promoting it [20]. The impact of not including this factor in clinical decision support systems and prediction tools is unknown, but it could not be negligible. The low quality of information regarding PA in health electronic records may have caused this scarcity of works that include PA as a covariate [19]. Nevertheless, ongoing improvements in the usability of clinical health records within the real-world data framework offer an opportunity to assess the influence of PA on relevant outcomes.

The aim of this study was to estimate to what extent having an active or partially active lifestyle can contribute to reducing or postponing the risk of fatal and nonfatal CVD events and all-cause mortality during a five-year follow-up among patients with T2D in a population-based cohort from a southern European country.

## 2. Materials and Methods

### 2.1. Study Population

The data used in this study belonged to the population-based CARDIANA cohort (CARdiovascular Risk in patients with DIAbetes in NAvarra), which includes all users of the Public Health Service of Navarra who, on 1 January 2012, had an active code of type 1 diabetes (T1D) or T2D (T89 and T90 of the International Classification of Primary Care, version 2 [ICPC-2], respectively) in the ATENEA records, which is the primary care electronic medical record system of Navarra. Navarra is an autonomous region in northern Spain with more than 600,000 inhabitants, of which more than half live in the metropolitan area of Pamplona, the capital of the province. It is the third out of the 17 autonomous regions of Spain in the ranking of highest mean income, and the second in the ranking of life expectancy (84.4 years). In this region, citizens are covered by the Regional Health Service of Navarra-Osasunbidea, which is part of the National Health System of Spain, and only 3.2% of the population has private or mixed health insurance [21]. Primary care electronic medical records were established in Navarra in the early 2000s and have been thoroughly used by all professionals since 2008.

The CARDIANA cohort takes information from several clinical and administrative databases. The primary care health record, ATENEA, is the core of the generated database and contains the main administrative and clinical baseline characteristics. The LAKORA-TIS dataset and the population register complemented previous administrative demographic information. LAMIA database provided pharmacologic information, and the HCI database provided information regarding the specialist health care units. The HIS-LEIRE contains the Minimum Basic Data Set (MBDS), with diagnosis and procedures on all hospital discharges coded according to the International Classification of Diseases: ICD-9CM until 31 December 2015 and ICD-10-ES from 1 January 2016-on. The mortality registry provided the date and cause of death according to ICD-10-ES classification system. Finally, the Type 1 diabetes registry provided the list of all type 1 diabetes patients with onset date after 31 December 1989. Thus, the final cohort contains anonymized patient-level information on socioeconomic, lifestyle-related variables and relevant CVD clinical risk factors at baseline and main outcomes and use of health services along with the evolution. The final cohort consisted of 1077 patients with T1D and 33,842 patients with T2D, and all participants were followed for 5 years, from 1 January 2012 to outcome occurrence or end of follow-up on 31 December 2016, except for those who moved to another community or country (*n* = 455, 1.3%) and who were censored the date they were deregistered. For the current study, we included men and women with prevalent T2D and available information about PA.

The study protocol for the creation of the cohort and assessment of CVD risk factors was favorably evaluated by the Ethics Committee of Clinical Research of Navarra (Project 2015/111), a session on 20 October 2015, and updated in Project 97/2019 session on 28 August 2019).

### 2.2. Exposure

Information regarding general PA was obtained from the primary care records. This variable is included in the ‘self-management protocol’ file, which was implemented in the region to promote vascular prevention among patients considered at risk, including among patients with type 2 diabetes. It was filled by nurse practitioners, and it has a drop-down menu that distinguishes four categories, namely disability in mobility, inactive, partially active or active. Patients with no data in this variable or with data that had a registration date previous to 1 January 2007 (five years before the study initiation) were considered to have missing data on this variable and were not included in the analyses. The category “disability in mobility” was included in the “inactive category” because of the very low number of patients (2% of the total valid). For patients with assigned free-text value not included in the pre-specified categories, a natural language process code was created ad hoc to assign them to one of the categories if the assignment could be considered unequivocal and were considered missing otherwise. This procedure recovered 6% of the data.

### 2.3. Outcomes

Two principal endpoints were considered along the follow-up, a composite endpoint including fatal and non-fatal CVD events and another endpoint including all-cause mortality. A CVD event was considered to occur when CVD diagnostic or procedure codes were recorded in the mortality registry or in the Minimum Basic DataSet (MBDS), which includes diagnosis and procedures on all hospital discharges coded according to the International Classification of Diseases (ICD) revisions 9 and 10. We used as reference the codes used in Read et al. [22], which included coronary heart diseases and cerebrovascular diseases (codes ICD10: I20–I25, I46, I60–I63, I65, G45, R96 and ICD9: 410, 430–435).

### 2.4. Confounders

Several known CVD risk factors or proxies were taken into account in the analysis as potential confounders for the analyses of the relationship between AF and CVD risk or mortality. These are sex, age, education level, time since T2D diagnosis, previous history of CVD (codes K74–K77, K89, K90 and K91 in the International Classification of Primary Care, version 2 [ICPC-2]) and comorbidity. Comorbidity was measured using an abbreviated version of the Charlson comorbidity index, aCharlson, which assigns one point to comorbidities CVD, diabetes, heart failure, pulmonary obstructive chronic disease (POCD), dementia and peripheral artery disease and two points for chronic renal failure (CRF) and cancer, therefore ranging from zero to ten points [23,24]. 

Other potential CVD risk factors, such as body mass index (BMI), smoking status, alcohol intake and hypertension (HTA), were also included. The presence of HTA was considered when participants had a systolic blood pressure ≥140 or a diastolic blood pressure ≥90 following actual clinical guidelines [25], and when this information was not available, HTA was considered present if the participant was under any treatment for HTA. Some important baseline laboratory test parameters, such as HbA1c, total cholesterol, albumin to creatinine ratio and LDL were also included in complementary analyses.

### 2.5. Statistical Analysis

Baseline characteristics were summarized by the PA group using descriptive measures, such as the means with standard deviations and frequencies with percentages. Two methodological approaches were conducted to account for important confounders. One had a cohort design and used survival methods for the analysis, and the other had a matched nested case-control study design and used logistic regression for the analysis.

#### 2.5.1. Cohort Design

Kaplan–Meier curves were used to visualize cumulative incidence by PA group for CVD events and overall survival. Complementarily, to estimate the magnitude of the effect on each outcome, univariate (Model 1) and multivariate Cox proportional hazards models were fitted using age as a time scale, as recommended in [26], and adjusted for age, sex, duration of diabetes, previous history of CVD and aCharlson (Model 2) and included other potential risk factors such as smoking, alcohol intake, BMI and HTA (Model 3). The Cox proportionality assumption was evaluated by testing the correlation between the corresponding scaled Schoenfeld residuals with time. Variables that did not meet this assumption were included as time-dependent coefficient variables in order to achieve the proportionality requirement. Factors used for model adjustment were based on an adapted directed acyclic graph (DAG) from Andersen et al. [27,28], including the aCharlson confounder, as it was considered to have an impact on both outcomes (CVD risk and mortality) and PA (Appendix A). According to this DAG, Model 2 and Model 3 included the minimal sufficient adjustment variables for estimating the total and direct effects of PA on CVD, respectively. No imputation was made for missing data.

#### 2.5.2. Nested Case–Control Design

Two independent nested case–control analyses were performed, one for each endpoint (CVD or all-cause mortality). The methodology was the same for both analyses. Cases were defined as subjects with an incident outcome from the start of the study onward, as in the cohort study. The matching method used was the incidence density sampling method. This means that all participants without an event at the time point when a case was diagnosed (index date) were considered possible controls. Among these, we randomly selected up to two controls and matched them with the index case by age category (±2 years), sex, study level, duration of T2D (±4 years), history of CVD and aCharlson category (aCharlson score: 1–2 points, 3–4 points or 5–8 points, as no one had more than 8 points). Only subjects with complete information at the education level (the only matching variable with missing data, 1.9%) were included in this analysis (*n* = 26,083). Univariate and multivariate conditional logistic regression models with PA as exposure and CVD events or mortality as outcome were fitted, including smoking status, alcohol intake, BMI and HTA as possible confounders, obtaining unadjusted and adjusted odds ratios with 95% confidence intervals.

#### 2.5.3. Complementary Analyses

Three complementary analyses were conducted to assess the stability of the results and possible interactions among variables. First, an additional Cox regression model was fitted with all variables in Model 3 plus the following baseline laboratory test parameters: HbA1c, cholesterol, albumin to creatinine ratio and LDL. Second, we assessed whether the effect of PA on CVD events and mortality differed with age by conducting subgroup analyses stratifying patients <65 years old and ≥65 years old. Finally, a subgroup analysis for the CVD outcome was conducted stratifying the cohort by history of CVD.

All analyses were performed in R 4.1.0 (R Foundation for Statistical Computing, Vienna, Austria; https://cran.r-project.org/src/base/R-4/, accessed on 2 November 2021). For survival modeling, libraries “survival” and “survminer” were used. For the matching method, library “Epi” was used.

## 3. Results

The baseline characteristics of the study participants at the start of the study are shown in Table 1. From the total prevalent T2D cohort with 33,842 patients, PA data were valid for 26,587 (78.6%) patients, and these were the patients who were included in the study cohort. The mean age of this cohort was 70.5 years, and 54.9% of them were males. Regarding PA, 12.6% belonged to the inactive category, 31.5% to the partially active category and 55.9% to the active category. People in the physically active lifestyle group were younger, had a lower BMI, lower aCharlson scores and higher educational level, had a higher probability of being male, smokers and alcohol users and had a lower probability of having HTA and a history of CVD.

Among all included participants, 2362 (8.9%) CVD cases were reported during the five-year follow-up (median follow-up equal to 28 months), with 11.6%, 10.1% and 7.6% in the inactive, partially active and active subgroups, respectively. A total of 5111 (19.2%) patients died by any cause (median follow-up equal to 30.4 months), with 37.8%, 23.9% and 12.4% in the inactive, partially active and active subgroups, respectively.

### 3.1. Cohort Design Results

Kaplan–Meier curves are shown in Figure 1, which indicate that the risk of CVD events and all-cause mortality decreased as the level of PA increased.

The crude and adjusted hazard ratios (HRs) of CVD events and mortality are shown in Table 2. In the univariate analysis, compared with the inactive PA group, participants in the partially active and active PA groups had HRs equal to 0.81 (95% CI: 0.72–0.92) and 0.65 (95% CI: 0.58–0.73) for CVD events and 0.65 (95% CI: 0.61–0.70) and 0.44 (95% CI: 0.41–0.47) for mortality, respectively. In the multivariate analyses, aCharlson was included as a time-dependent coefficient variable since it showed a remarkable time-scale dependency. Five different coefficients were modeled for the aCharlson variables, one for each of the following age ranges: <55 yrs, 55–65 yrs, 65–75 yrs and >75 yrs (Appendix A). After adjusting for the minimum set of variables that included sex, age, education level, duration of T2D, history of CVD and time-dependent coefficient aCharlson (Model 2), the effect of PA on both outcomes remained significant, with a 7–9% higher HR in the case of CVD event and similar in magnitude in the case of mortality. Finally, Model 3, which included additional potential risk factors (such as smoking status, alcohol intake, BMI and HTA) provides an HR for the active versus inactive PA group equal to 0.72 (95% CI: 0.61–0.84) in the CVD event assessment and equal to 0.51 (95% CI: 0.46–0.57) in the mortality assessment. In this last model, the proportional hazards requirement was achieved for all variables. Complete regression models (Model 3) are shown in Appendix A.

### 3.2. Nested Case–Control Analysis Results

The incidence density matching technique applied to the cohort of 26,083 patients with no missing data for the matching variables on the CVD outcomes created a matched set of 2104 cases and 4032 controls after excluding 221 cases for which no control could be found. Among the case–control sets, 176 sets were incomplete, that is, they had fewer than four controls. On the other hand, this matching technique applied for the outcome defined by all-cause mortality events provided a matched set of 4583 cases and 8727 controls after excluding 460 cases for which no control was found. Among these case–control sets, 439 sets were incomplete.

Table 3 shows baseline characteristics of cases and controls for both analyses. For the CVD outcome, all variables used in the matching procedure were balanced, whereas for the remaining variables, only smoking status, HTA and PA showed differences, with a higher proportion of smokers, patients with HTA and inactive patients among the group of cases. For the all-cause mortality outcome, all variables used in the matching procedure were balanced and differences between groups only existed in smoking status, alcohol consumption, HTA and PA, with the group of cases having a higher proportion of smokers, hypertensive patients and inactive patients but a lower proportion of alcohol users and lower figures of BMI.

The crude and adjusted odds ratios (ORs) of CVD events and mortality are shown in Table 4. Compared with the inactive group of patients, those in the partially active and active groups showed a lower risk of having a CVD event, with adjusted ORs equal to 0.84 (0.66–1.07, *p* = 0.160) and 0.71 (0.56–0.91, *p* < 0.008), respectively. These two groups also showed a lower risk of mortality, with adjusted ORs equal to 0.72 (0.61–0.85, *p* < 0.001) for the partially active group and 0.50 (0.42–0.59, *p* < 0.001) for the active group.

### 3.3. Complementary Analyses

The addition of laboratory test parameters (HbA1c, total cholesterol, LDL and albumin to creatinine ratio) to Cox regression Model 3, which included age, study level, duration of T2D, history of CVD, smoking status, alcohol intake, BMI, HTA and PA, did not particularly change the HR estimates (data not shown).

Finally, the stratified analyses conducted for patients aged <65 years and patients ≥65 years did not differ (data not shown). The stratified analysis conducted on the subgroups of patients by history of CVD under the nested case–control design is given in the Appendix A. Appendix A contain the characteristics of cases and controls for the CVD and mortality outcomes, respectively. They showed that there were no differences among groups regarding matching variables in any of the subgroups. Estimates for the effect of PA on both outcomes are given in Appendix A. For the group of patients who had no CVD history before baseline and compared to those in the inactive group, those in the active group had adjusted ORs equal to 0.76 (95% CI: 0.53–1.08) and 0.52 (95% CI: 0.41–0.65) for CVD events and all-cause mortality outcomes, respectively. Similarly, for the group of patients with a CVD history, the adjusted OR for the active group was equal to 0.62 (95% CI: 0.45–0.87) and 0.46 (95% CI: 0.36–0.58) for the CVD event and all-cause mortality outcomes, respectively.

## 4. Discussion

This population-based cohort study conducted on all subjects with T2D diagnosed before 2012 in a northern autonomous community of Spain shows that, compared to the patients in the non-physically active group, patients in the active group showed a significant decrease of 30% in the risk of suffering a CVD event and a 50% decrease in all-cause mortality. Partial PA was also associated with a relevant 15% reduction in the risk of CVD and 25% in mortality that only reached statistical significance for mortality. These results remained consistent after adjustment for potential confounding factors, independent of the methodological approach used to account for confounders (nested case–control design or survival modeling applied to the total cohort) and when the subgroup analysis was conducted to assess the effect in the subpopulation with and without a CVD history, with a slightly more marked gradient in the group of patients with CVD history.

It has been estimated that approximately 23% of the adult population around the world and, more worrying, approximately 81% of teenagers do not meet the WHO global recommended levels of PA [20]. Screen time is a significant contributing factor to this sedentary behavior, which has undoubtedly worsened during the COVID-19 pandemic, when confinement, mobility restrictions, social distance and telework have been implemented [29]. Physical inactivity is a serious public health problem, especially in developed countries. It is considered to contribute substantially to mortality through the development of other important risk factors and metabolic-related chronic diseases, such as obesity, HTA, diabetes and CVD [20].

Evidence exists about the benefits of exercise in improving CVD risk and reducing all-cause mortality in the general population. In a prospective cohort of 130,000 middle-aged participants from 17 countries with different income levels, Lear et al. found that participants who performed moderate and high PA, recreational or not recreational, had approximately 15% and 25% reductions in major CVD events and approximately 20% and 35% reductions in total mortality, respectively, in comparison with participants in the low PA group [2]. Moreover, in a meta-analysis including 850,000 participants, Ekelund et al. showed that moderate and vigorous PA reduced the association between sitting time or TV-watching time with CVD and cancer-related mortality [4]. This is of great importance, especially for those who have sitting jobs. Since CVD is one of the most common causes of morbidity and mortality among patients with T2D [1], it could be hypothesized that PA could have similar or even higher protective effects in this risk population. In this regard, many researchers have conducted RCTs in patients with T2D to assess the effects of different exercise modalities (aerobic, resistance, strength or combinations) on the levels of classic CVD risk factors, including HbA1c, LDL, cholesterol, heart rate and weight. Among others, a meta-analysis assessing the impact of walking on risk factors in these patients [7] supported that walking decreased HbA1c, BMI and DBP levels. Furthermore, some exercise modalities could be more effective than others in regulating the levels of specific risk factors. A meta-analysis conducted to assess the comparative impact of different exercise training on risk factors [8] concluded that, compared to supervised aerobic or resistance exercise alone, combined exercise was more effective at reducing HbA1c, but no differences between modalities were found for other risk factors. Nevertheless, most studies agree that all types of PA improve the levels of the analyzed markers when compared with no exercise [9,10,14] and point out that using one or the other type of exercise may be less important than performing some form of PA [10].

Although the aforementioned studies provide sufficient evidence on the beneficial effects of short-term exercise interventions on specific CVD risk factors and surrogate markers, studies assessing the long-lasting beneficial effects of PA on fatal and nonfatal CVD clinical outcomes and all-cause mortality are not abundant. In a cohort of 270,000 adults with T2D in Sweden, with a median follow-up of 5.7 years, Rawshani et al. [16] found that a low level of PA was the second most important risk factor in terms of estimating relative risk for predicting all-cause mortality, only behind smoking; it ranked between third and fourth for predicting CVD-related events (acute myocardial infarction, stroke and heart failure), only after HbA1c, SBP, LDL and smoking. In another prospective study including 5859 patients with T2D, with a median follow-up of 9.4 years, Sluik et al. [18] found that, compared with participants in the inactive category, those in the active category had approximately 25% lower total mortality, and it was reduced to 20% when excluding patients with previous CVD events, cancer or less than two years of follow-up. In a further meta-analysis that included this and four other studies, a pooled 40% reduction in all-cause mortality was obtained. In our study, we observed large effect magnitudes for the active group, reaching a 50% reduction on the all-cause mortality in both, the total cohort and when selecting those patients without a history of CVD. In this sense, our results are in line with the meta-analysis findings.

This study has several strengths. First, the real-world database used contains detailed clinical, socioeconomic and behavioral information at the individual level and allows for a large sample size from a wide age range of patients, providing generalizable conclusions. Second, the realization of complementary methodological approaches that included cohort and nested matched case–control designs, together with the consistent findings results obtained from both designs and the complementary analyses supports the reliability of the results. It also has several limitations. First, as in most real-world data studies, the possibility of low-quality information for some variables and the presence of missing data, especially in those variables depending on the physician’s manual reporting, such as PA, can result in the presence of information bias. In this regard, to minimize missing information without inducing bias for the PA exposure variable, a date filter was included to ignore old stale data, and at the same time, we tried to use nonstructured information on this variable using text codes. Second, other important factors that could act as confounders or mediators between PA and health outcomes (CVD and mortality) have not been included. Among them, other lifestyle factors such as a healthy diet, sleeping quality and stress levels as well as other health factors that may limit the ability to be physically active were not included in this study due to insufficient information in medical records. Third, people with undiagnosed T2D or those using exclusively private health institutions were not included. Finally, some patients who were assumed to be followed during the whole period may actually be lost to follow-up because of having moved to another region.

Despite the clear role of PA on CVD risk, a recent systematic review on CVD risk prediction models applicable to patients with diabetes [19] has evidenced the lack of exercise-related variables in most widely used models. In fact, only one of the 19 models derived for patients with T2D [30] and six of the 46 models derived for the general population that have diabetes as a covariate [30,31,32,33,34,35] included PA as a covariate. This could be due to a lack of information regarding this modifiable factor in cohorts generated under the real-world data framework or to inadequate quality of the data. However, given the importance of PA in the development of CVD-related complications in patients with T2D, efforts should be made to register well-defined, standardized and complete information regarding this relevant lifestyle factor in electronic medical records, as more than one decade has passed since the implementation of electronic health records has taken place. Furthermore, taking into account that clinical guidelines for the management of patients with T2D recommend the use of prediction models as treatment decision aid tools, future research could be focused on developing and validating new prediction models in patients with T2D that include PA as a risk factor to assess the impact of implementing them on relevant CVD outcomes.

## 5. Conclusions

This population-based study shows that subjects with T2D who have a physically active lifestyle have an approximately 30% reduction in the risk of suffering a CVD event and an approximately 50% reduction in the risk of all-cause death. These results provide enhanced knowledge of the benefits of regular PA in patients with prevalent T2D.

## Figures and Tables

**Figure 1 ijerph-18-12370-f001:**
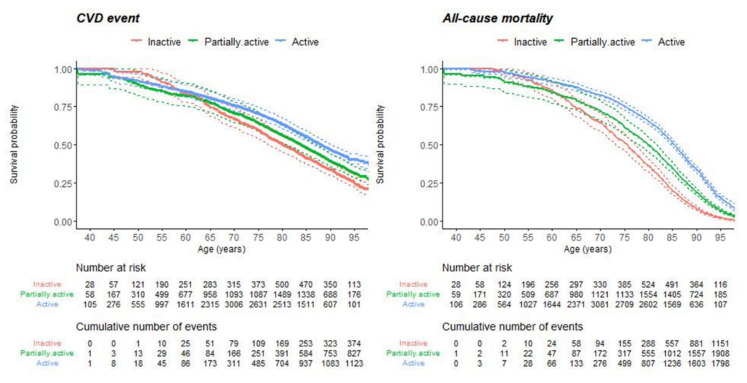
Kaplan–Meier curves by Physical Activity groups.

**Table 1 ijerph-18-12370-t001:** Baseline characteristics according to physical activity among subjects with Type 2 Diabetes.

Variable	Total *n* (%)	Levels	Inactive	Partially Active	Active
Total *n* (%)	26,587 (100.0)		3362 (12.6)	8371 (31.5)	14,854 (55.9)
Age	26,587 (100.0)	Mean (SD)	74.1 (13.5)	72.1 (12.5)	68.7 (11.1)
Sex	26,587 (100.0)	Male	1431 (42.6)	3966 (47.4)	9207 (62.0)
		Female	1931 (57.4)	4405 (52.6)	5647 (38.0)
BMI	19,753 (74.3)	Mean (SD)	32.9 (7.3)	31.2 (6.0)	29.8 (5.2)
Smoking status	25,395 (95.5)	Non-smoker	2162 (67.4)	5147 (64.1)	7981 (56.4)
		Ex-smoker	515 (16.0)	1614 (20.1)	3774 (26.7)
		Smoker	533 (16.6)	1266 (15.8)	2403 (17.0)
Alcohol	24,843 (93.4)	No	2322 (75.1)	5547 (70.6)	8740 (62.9)
		Yes	770 (24.9)	2308 (29.4)	5156 (37.1)
Duration T2D	26,587 (100.0)	Median(IQR)	7.7 (4.4–11.9)	7.6 (4.1–11.6)	7.1 (3.8–10.8)
Education	26,083 (98.1)	No	1418 (43.0)	3170 (38.6)	4737 (32.5)
Level		Primary School	1553 (47.1)	4179 (50.9)	7952 (54.5)
		High School	203 (6.2)	575 (7.0)	1287 (8.8)
		University level	123 (3.7)	284 (3.5)	602 (4.1)
HDL	20,354 (76.6)	Mean (SD)	46.2 (13.7)	48.8 (13.9)	49.8 (14.1)
LDL	19,896 (74.8)	Mean (SD)	108.4 (33.6)	110.2 (32.3)	110.7 (30.9)
TGC	20,231 (76.1)	Median (IQR)	135.0 (98–186)	130.0 (95–179)	120.0 (87–167)
SBP	23,558 (88.6)	Mean (SD)	75.4 (11.4)	75.8 (10.5)	76.4 (10.2)
DBP	23,560 (88.6)	Mean (SD)	135.4 (18.3)	136.4 (17.8)	135.4 (16.5)
HTA treatment	26,587 (100.0)	No	1419 (42.2)	3612 (43.1)	5649 (38.0)
		Yes	1943 (57.8)	4759 (56.9)	9205 (62.0)
HTA	26,587 (100.0)	No	1308 (38.9)	3014 (36.0)	6060 (40.8)
		Yes	2054 (61.1)	5357 (64.0)	8794 (59.2)
Albumin/creatinine	16,225 (61.0)	Median (IQR)	9.6 (4.0–30.1)	8.0 (3.9–23.0)	6.0 (3.0–14.9)
Total Chol	21,713 (81.7)	Median (IQR)	182.0(155–210)	186.0 (161–212)	185.0 (162–210)
Cardiac freq.	20,441 (76.9)	Mean (SD)	74.0 (66.0–82.0)	72.0 (65.0–80.0)	72.0 (64.0–80.0)
HbA1c	18,553 (69.8)	Median (IQR)	6.8 (6.2–7.9)	6.8 (6.2–7.7)	6.7 (6.1–7.5)
History of CVD	26,587 (100.0)	No	2178 (64.8)	6053 (72.3)	11,783 (79.3)
		Yes	1184 (35.2)	2318 (27.7)	3071 (20.7)
aCharlson	26,587 (100.0)	1–2	1739 (51.7)	4760 (56.9)	9427 (63.5)
		3–4	1243 (37.0)	2879 (34.4)	4576 (30.8)
		5–8	355 (10.6)	674 (8.1)	796 (5.4)

BMI: Body mass index (kg/m^2^). T2D: Diabetes mellitus; DBP: Systolic blood pressure (mm Hg); SBP: Diastolic blood pressure (mm Hg); HTA: Hypertension; HDL: High density lipoprotein (mg/dL); LDL: Low density lipoprotein (mg/dL): TGC: Triglycerides (mg/dL); Total Chol: Cholesterol total (mg/dL); Cardiac freq: Cardiac frequency (bpm); HbA1c: Glycosylated hemoglobin (%); aCharlson: abbreviated Charlson comorbidity index.

**Table 2 ijerph-18-12370-t002:** Unadjusted and adjusted HRs (95% CI) of CVD and mortality according to physical activity among subjects with Type 2 diabetes.

Models	Units	CVD	Mortality
^1^ Model 1	Inactive	Ref.	Ref.
	Partially active	0.81 (0.72–0.92, *p* = 0.001)	0.65 (0.61–0.70, *p* < 0.001)
	Active	0.65 (0.58–0.73, *p* < 0.001)	0.44 (0.41–0.47, *p* < 0.001)
^2^ Model 2	Inactive	Ref.	Ref.
	Partially active	0.88 (0.78–1.00, *p* = 0.043)	0.67 (0.63–0.72, *p* < 0.001)
	Active	0.74 (0.65–0.83, *p* < 0.001)	0.45 (0.42–0.49, *p* < 0.001)
^3^ Model 3	Inactive	Ref.	Ref.
	Partially active	0.87 (0.74–1.03, *p* = 0.097)	0.75 (0.67–0.83, *p* < 0.001)
	Active	0.72 (0.61–0.84, *p* < 0.001)	0.51 (0.46–0.57, *p* < 0.001)

^1^ Model 1 covariates: Physical activity (PA); ^2^ Model 2 covariates: sex, age, study level, duration of T2D, history of CVD, aCharlson and PA; ^3^ Model 3 covariates: variables in Model 2 plus smoking status, alcohol intake, body mass index and hypertension.

**Table 3 ijerph-18-12370-t003:** Characteristics of the CVD group and the control group included in nested case–control study from the total cohort of subjects with Type 2 diabetes.

		CVD			All-Cause Mortality		
Variable	Levels	Controls	Cases	*p*-Value ^δ^	Controls	Cases	*p*-Value ^δ^
Total *n* (%)		4032 (65.7)	2104 (34.3)		8727 (65.6)	4583 (34.4)	
Age	Mean (SD)	75.0 (9.9)	75.0 (10.1)	0.890	79.4 (9.0)	79.5 (9.1)	0.474
Sex	Male	2402 (59.6)	1255 (59.6)	0.977	4685 (53.7)	2469 (53.9)	0.850
	Female	1630 (40.4)	849 (40.4)		4042 (46.3)	2114 (46.1)	
Education	No	1632 (40.5)	847 (40.3)	0.505	4133 (47.4)	2161 (47.2)	0.300
Level	Primary School	2171 (53.8)	1123 (53.4)		4268 (48.9)	2218 (48.4)	
	High School	176 (4.4)	101 (4.8)		210 (2.4)	131 (2.9)	
	University	53 (1.3)	33 (1.6)		116 (1.3)	73 (1.6)	
Duration T2D	Median (IQR)	8.0 (5.0–12.0)	9.0 (6.0 to 12.0)	0.219	9.0 (6.0–13.0)	9.0 (6.0–13.0)	0.468
History of CVD	No	2195 (54.4)	1128 (53.6)	0.555	5063 (58.0)	2621 (57.2)	0.369
	Yes	1837 (45.6)	976 (46.4)		3664 (42.0)	1962 (42.8)	
Smoking	Non-smoker	2368 (61.3)	1197 (59.9)	0.001	5767 (68.4)	2884 (66.0)	<0.001
	Ex-smoker	1009 (26.1)	481 (24.1)		1894 (22.5)	968 (22.1)	
	Smoker	489 (12.6)	319 (16.0)		767 (9.1)	519 (11.9)	
Alcohol	No	2533 (66.3)	1361 (68.4)	0.116	5882 (71.0)	3174 (73.7)	0.002
	Yes	1285 (33.7)	628 (31.6)		2402 (29.0)	1133 (26.3)	
BMI	Mean (SD)	30.2 (5.5)	29.9 (5.1)	0.086	29.6 (5.4)	29.4 (6.0)	0.108
HTA	No	901 (22.3)	413 (19.6)	0.015	1790 (20.5)	1075 (23.5)	<0.001
	Yes	3131 (77.7)	1691 (80.4)		6937 (79.5)	3508 (76.5)	
aCharlson	1–2	1569 (38.9)	803 (38.2)	0.286	2848 (32.6)	1473 (32.1)	0.174
	3–4	1856 (46.0)	957 (45.5)		4616 (52.9)	2393 (52.2)	
	5–6	607 (15.1)	344 (16.3)		1263 (14.5)	717 (15.6)	
PA	Inactive	518 (12.8)	340 (16.2)	<0.001	1336 (15.3)	1128 (24.6)	<0.001
	Partially active	1376 (34.1)	749 (35.6)		3174 (36.4)	1779 (38.8)	
	Active	2138 (53.0)	1015 (48.2)		4217 (48.3)	1676 (36.6)	

^δ^ X^2^ test used for comparison except for study level, aCharlson and Physical activity (PA), for which test for trend in proportions was used. BMI: body mass index, HTA: hypertension; aCharlson: abbreviated Charlson comorbidity index; PA: Physical activity.

**Table 4 ijerph-18-12370-t004:** Unadjusted and adjusted ORs (95% CI) of CVD and mortality according to physical activity among subjects with Type 2 diabetes in the case–control study.

Models	Units	CVD	Mortality
^1^ Model 1	Inactive	Ref.	Ref.
	Partially active	0.84 (0.71–0.99, *p* = 0.033)	0.65 (0.58–0.71, *p* < 0.001)
	Active	0.72 (0.61–0.84, *p* < 0.001)	0.43 (0.39–0.48, *p* < 0.001)
^2^ Model 2	Inactive	Ref.	Ref.
	Partially active	0.84 (0.66–1.07, *p* = 0.160)	0.72 (0.61–0.85, *p* < 0.001)
	Active	0.71 (0.56–0.91, *p* < 0.008)	0.50 (0.42–0.59, *p* < 0.001)

^1^ Model1 covariates: Physical Activity (PA); ^2^ Model2 covariates: smoking status, alcohol intake, body mass index, hypertension and PA.

## Data Availability

Restrictions apply to the availability of these data. Data were obtained from the Navarra Health System, Osasunbidea, and the NASTAT institutions. Data requests will only be considered after the approval of the research ethics committee from the solicitor institution and also from Osasunbidea and NASTAT institutions, who are responsible for the clinical information and the population information, respectively.

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
