# Peer review of "Effect of Physical Activity on Cardiovascular Event Risk in a Population-Based Cohort of Patients with Type 2 Diabetes"

_ijerph, 2021, doi:10.3390/ijerph182312370_

Round 1

Reviewer 1 Report

Complex analysis of the data.  Great work! 

Reviewer 2 Report

Effect of Physical Activity on Cardiovascular Event Risk in a 2 Population-Based Cohort of Patients with Type 2 Diabetes.

This retrospective study utilised a population cohort of 26,587 patients with Type 2 diabetes from the Navarre health system, which were followed for 5 years to estimate the extend physical activity can reduce cardiovascular disease risk and all-cause mortality in patients with Type 2 diabetes. Cardiovascular disease is the most common cause of morbidity and mortality among patients with Type 2 diabetes, and participation in regular physical activity has proven advantages of on reducing cardiovascular disease. This is an interesting study, which is also well-written.

I have few minor queries:

Abstract

  1. Line 27: “The aim of this study is to...” Should this be ‘is’ or ‘was’?

  1. Lines 33-34: “A total of 5111 (19.2%) patients died during follow-up, 37.8, 23.9 and 12.4% of the inactive, partially active and active groups, respectively.” The latter part of the sentence is not clear. The inactive, partially active and active groups do not connect well with ‘patients died during follow-up’. Also, put sign of percentage in respect of all two figures (37.8, 23.9, 6.5, 5.6).

  1. Line 40: I suggest the sentence should read: “This study provides further evidence that physical active patients with T2D may have reduced risk of CVD related complications and all-cause mortality”.

Introduction

  1. Line 70: “this lack of works”. This is not clear

Results

  1. Lines 188 & 190: Insert the sign of percentages for all the figures.

  1. Line 208: “In this last model, the proportional hazards requirement is met for all variables.”. I suggest you change the sentence to read: “In this last model, the proportional hazards requirement was achieved for all variables.”

Discussion

  1. Line 306: “This is of great importance, especially for those who have sedentary occupations.” What is sedentary occupations mean?

Reviewer 3 Report

This paper addresses an interesting and important issue. But there are some serious flaws or weaknesses. 

Firstly, the rationale for this study is unclear. Why are we studying this particular group of people who already have T2 Diabetes, and potential CVD issues? Why is it only restricted to diabetics? If you have all of these linked medical records, why not study physical activity and health outcomes amongst all people. This is not explained at all, except for saying many diabetics have CVD as an co-morbidity. 

The Methods lack many key details. To start with what is the study setting is not explained. Many readers may not know what or where Navarra is. What country is it in? Is it urban or rural? Large or small population? High or low income? Ageing population or young? These are important for interpreting the results. 

There is no detail about how the physical activity levels were measured except that they came from primary health care medical records. How were these data collected? What criteria was used to judge PA as inactive or active. This is really the crucial variable in this study and i am not confident based on the description that it is robust enough to serve its purposes in this paper. 

There is no detail on other data aspects either. For example diabetics were identified from primary health care records. How then were mortality cases, and new CVD events identified? Is there data linkage going on? The authors do not explain. 

Finally, i am not convinced by the underlying premise of the analysis. Physical activity measured up to 5 years before identification as diabetic is tricky to directly link to following CVD events or all cause mortality. There are many health factors which may limit the ability to be physically active and these may also increase the risk of all cause mortality. So really being measured here is the relationship between physical activity and all cause mortality. CVD and diabetes may not be playing a major role here. 

Round 2

Reviewer 3 Report

I thank and commend the authors for carrying out this extensive revision. You have really added the elements that i thought were missing. I now support the paper being accepted.